# Two-Dimensional Transition Metal-Hexaaminobenzene Monolayer Single-Atom Catalyst for Electrocatalytic Carbon Dioxide Reduction

**DOI:** 10.3390/nano12224005

**Published:** 2022-11-14

**Authors:** Xianshi Zeng, Zongxing Tu, Yanli Yuan, Luliang Liao, Chuncai Xiao, Yufeng Wen, Kai Xiong

**Affiliations:** 1Institute for Advanced Study, School of Physics and Materials Science, Nanchang University, Nanchang 330031, China; 2School of Chemistry and Chemical Engineering, Nanchang University, Nanchang 330031, China; 3School of Mechanical and Electrical Engineering, Xinyu University, Xinyu 338004, China; 4School of Mathematical Sciences and Physics, Jinggangshan University, Ji’an 343009, China; 5Materials Genome Institute, National Center for International Research on Photoelectric and Energy Materials, School of Materials and Energy, Yunnan University, Kunming 650091, China; 6Advanced Computing Center, Information Technology Center, Yunnan University, Kunming 650091, China

**Keywords:** CO_2_ reduction reaction, electro-catalysis, single-atom catalysts, two-dimensional materials, transition metal-hexaaminobenzene, density functional theory (DFT) calculations

## Abstract

Electrocatalytic reduction of CO_2_ to valuable fuels and chemicals can not only alleviate the energy crisis but also improve the atmospheric environment. The key is to develop electrocatalysts that are extremely stable, efficient, selective, and reasonably priced. In this study, spin-polarized density function theory (DFT) calculations were used to comprehensively examine the catalytic efficacy of transition metal-hexaaminobenzene (TM-HAB) monolayers as single-atom catalysts for the electroreduction of CO_2_. In the modified two-dimensional TM-HAB monolayer, our findings demonstrate that the binding of individual metal atoms to HAB can be strong enough for the atoms to be evenly disseminated and immobilized. In light of the conflicting hydrogen evolution processes, TM-HAB effectively inhibits hydrogen evolution. CH_4_ dominates the reduction byproducts of Sc, Ti, V, Cr, and Cu. HCOOH makes up the majority of Zn’s reduction products. Co’s primary reduction products are CH_3_OH and CH_4_, whereas Mn and Fe’s primary reduction products are HCHO, CH_3_OH, and CH_4_. Among these, the Ti-HAB reduction products have a 1.14 eV limiting potential and a 1.31 V overpotential. The other monolayers have relatively low overpotentials between 0.01 V and 0.7 V; therefore, we predict that TM-HAB monolayers will exhibit strong catalytic activity in the electrocatalytic reduction of CO_2_, making them promising electrocatalysts for CO_2_ reduction.

## 1. Introduction

In recent years, a large amount of CO_2_ has been released into the atmosphere, leading to a series of environmental and socio-economic problems. Converting CO_2_ into hydrocarbon fuels and valuable chemical raw materials such as methane, methanol, and formic acid is important for resource utilization and environmental protection, as well as for global sustainable development [1,2,3,4,5,6,7]. However, it is not easy to achieve efficient electrocatalytic CO_2_ reduction, which is a challenge because the carbon atoms in the CO_2_ molecule adopt sp hybridization to bond with two oxygen atoms to form C=O bonds, which endows the CO_2_ molecule with high chemical inertness and low solubility in aqueous solutions. Currently, CO_2_ conversion can be carried out using different catalytic approaches involving chemical [8], photochemical [9], electrochemical [10], and biological methods [11]. Among the various methods, electrochemical methods have been widely noted for their mild conditions, simple operation, safety and stability, and easy access to intermittent energy sources such as solar and wind power [12]. In the electrocatalytic CO_2_ reduction process, CO_2_ molecules are activated by the interaction with the catalyst surface under the action of external electrical energy, and the subsequent reaction steps can be carried out further under mild conditions with relatively low energy cost. However, CO_2_ activation by high-activity electrocatalysts, the products of CO_2_RR, generally occurs with low selectivity. Therefore, it is quite urgent to design and fabricate catalysts with high activity and distinct selectivity. It is well known that effective activation of carbon dioxide requires electron transfer from the substrate to the carbon dioxide molecule, which leads to appropriate accompanying structural deformation [13,14,15,16,17,18]. In the past few years, various heterogeneous catalysts for the activation of inert CO_2_ have been proposed, including pure metals [19], metal oxide interfaces [20], graphite-based materials [21], sulfides [22], and metal organic frameworks (MOFs) [23]. To meet the high activity and selectivity for specific products, researchers have combined the aforementioned heterogeneous catalysts with the emerging nanoscience for functional applications [24]. Supported metal nanoparticles have been a widespread approach in recent years, by which to reshape heterogeneous catalysts to meet different needs. The size of metal particles is an important factor affecting catalytic performance. As the size of metal particles decreases, their surface area/volume ratio and catalytic activity can be significantly improved. Rationally designed single-atom catalysts (SACs) for CO_2_ conversion show high activity and significant selectivity. The reduction of CO_2_ to CH_3_OH is catalyzed by $\beta_{12}$ boron monolayers supported by V atoms [25], as well as the conversion of CO_2_ to CH_4_ on the surface of SACs fabricated from porphyrin-like graphene supported by single Co, Rh, and Ir atoms [26]. In addition to the type of metal atoms supported on the surface of nanomaterials, organometallic complexes anchored on substances, as well as anchored on porous materials involving MOFs, zeolites, and other ion-exchange metals on the surface, can also be considered as SACs [27,28]. Thus, MOFs that include low-coordination metal centers individually and regularly dispersed on the surface are promising for a wide range of SAC applications in the near future [29,30,31,32,33,34,35,36,37,38,39,40,41,42].

Two-dimensional MOFs may be directly synthesized without the need for material modification to become highly useful SACs. Additionally, a variety of metals may be used for the metal centers of two-dimensional MOFs, which always serve as active sites and impact the performance of the catalyst. By changing the kind of metal center to suit our demands, we may make use of the tunability of 2D MOFs. Additionally, MOFs possess metallic characteristics that guarantee quick electron transfer during electrochemical catalysis, a crucial quality of effective electrocatalysts. The formation of TMN_4_ complexes, which have high catalytic characteristics comparable to those of noble metals in a variety of catalytic interactions such oxygen reduction, nitrogen fixation, and CO_2_ reduction, is noteworthy [43,44,45,46]. Recently, Louie’s group successfully fabricated a series of novel two-dimensional MOFs using a bottom-up technique in which the transition metal (TM) atoms serve as the metal center and hexamethylbenzene (HAB) serves as the ligand [47]. Low-coordination TM atoms as metal centers are regularly and stably immobilized to form MOFs with the molecular formula TM_3_(HAB)${}_{2}$. The low-coordinated TM atoms are stably anchored as metal centers in the MOF TM_3_(HAB)${}_{2}$, exhibiting the characteristics of single-atom catalysts (SACs). Therefore, they can be considered SACs with high practicality. The metal center in the two-dimensional TM-HAB plays the role of an active site with catalytic properties, and the type of central atom can be tuned to meet the catalytic requirements [29]. Notably, TM-HAB is formed by each TM atom with four surrounding N atoms. The TMN_4_ complex is an analogue of TMNx, which exhibits good catalytic properties similar to those of noble metals in various catalysts, such as in oxygen reduction, nitrogen fixation, and carbon dioxide reduction [43,44,45,46]. Park et al. [48] proposed a two-dimensional (2D) conductive metal-organic framework consisting of M-N_4_ units (M = Ni, Cu) and hexaaminobenzene (HAB) linkers as a catalyst for oxygen reduction reactions, and the results showed that the catalytic performance depends strongly on the metal species. However, the application of TM-HAB monolayers for CO_2_ reduction has been little reported so far. This motivated us to explore whether TM-HAB constructed with different metal species could be used as a prospective electrocatalyst for CO_2_ reduction. In this paper, the catalytic properties of the first periodic transition-metal series TM-HAB monolayers for CO_2_ were systematically investigated using spin-polarized density function theory (DFT). Our results show that most TM-HAB monolayers exhibit excellent catalytic activity and strong stability. Therefore, we predict that TM-HAB monolayers can contribute to the next generation of low-cost, high-stability electroreduction catalysts.

## 2. Computational Methods

The Dmol3 package’s spin-polarized density functional theory (DFT) [49] was used for all computations in this work. The Perdew-Burke-Ernzerhof (PBE) function of the generalized gradient approximation (GGA) was used to accomplish the exchange correlation of electrons [50]. Through Grimme’s technique, which has been employed for different gas adsorptions and gas phase catalysts, the van der Waals (vdW) force was used to deal with long-range dispersion correction [51,52,53,54]. For transition metals, the density functional semicore pseudopotential (DSPP) was used to properly calculate other electrons and substitute core electrons with an effective pseudopotential [55]. The double numerical (DN) function was selected as the basis set for other atoms. The periodic supercell of each MOF contains 12 hydrogen atoms, 12 nitrogen atoms, 12 carbon atoms, and 3 transition metal atoms, and a unit cell of 14.33 × 14.33 × 25 Å^3^ with a vacuum space of 25 Å along the Z axis, which is spacious enough to avoid nonphysical interaction with the periodic image. The Brillouin zone (BZ) k-point sample was a 5 × 5 × 1 grid with a Monkhorst–Pack design [56]. The convergence thresholds for energy, gradient, and displacement for all geometric optimizations were 1.0 × 10^−5^ Ha, 2.0 × 10^−3^ Ha/Å, and 5.0 × 10^−3^ Å, respectively. The conductor-like screening model (COSMO) was used to simulate the electrolyte, which is always an aqueous environment, with a dielectric constant of 78.54 for H_2_O as the solvent [57]. The sorption energy (Eads) of CO_2_ on the TM-HAB periodical unilayer is defined as:(1)$$E_{\mathrm{ads}}=E_{{\mathrm{TM}-\mathrm{HAB}-\mathrm{CO}}_{2}}-E_{\mathrm{TM}-\mathrm{HAB}}-E_{{\mathrm{CO}}_{2}}$$
where $E_{{\mathrm{TM}-\mathrm{HAB}-\mathrm{CO}}_{2}}$, $E_{\mathrm{TM}-\mathrm{HAB}}$, and $E_{{\mathrm{CO}}_{2}}$ are the total energy for the CO_2_ adsorbed on the TM-HAB monolayer, the pure TM-HAB monolayer, and the CO_2_ molecule, respectively. The computational hydrogen electrode (CHE) model proposed by Nrskov and coworkers to account for the energy of a proton–electron pair in aqueous solution was used to estimate the change in Gibbs free energy (ΔG) for each of the CO_2_RR steps [58,59,60]. Gibbs free energy is defined as:(2)$$\Delta G=\Delta E+\Delta E_{\mathrm{ZEP}}-T\Delta S+\Delta G_{\mathrm{pH}}+\Delta G_{U}$$
where ΔE is the response energy, which can be obtained directly from the DFT calculation. ΔE_ZPE_ is the zero-point change in energy, ΔS is the change in the entropy, and T is the systematic temperature (298.15 K). E_ZPE_ and S of the CO_2_RR interstitial are calculated from the vibrational frequency. $\Delta G_{\mathrm{pH}}=2.303\;k_{\mathrm{BT}\;\mathrm{pH}}\;$ is a modification of the free energy due to the variation in H^+^ concentration. pH is taken to be zero for acidic mediators in this article. ΔU = −neU, where n is the number of transferred electrons, e is the electronic charge and U is the applied voltage. The limiting potential (U_L_) of the CO_2_RR can be obtained from the free potential change ($\Delta G_{\mathrm{Max}}$) by the relation $U_{L}=-\Delta G_{\mathrm{Max}}/\mathrm{ne}$. The overpotential (η) is evaluated as the difference between the balanced potential and the limiting potential. Therefore, the overpotential is defined as:(3)$$\eta =U{}_{\mathrm{equilibrium}}-U{}_{L}$$

## 3. Results and Discussion

### 3.1. Structural Features and Properties of the TM-HAB Monolayer

The optimized architectures of the TM-HAB materials are depicted in Figure 1. Appendix A lists specific lattice constant values. Each MOF has three transitional metal atoms, 12 hydrogen atoms, 12 nitrogen atoms, and 12 carbon atoms in each periodic cell. The lattice parameters of each cell are shown in Appendix A. Figure 1 displays models of a periodic 2 × 2 supercell. Each transition metal atom binds to two HAB molecules in the TM-HAB monolayer. The atoms in each of the 10 transition metals that we took into consideration (from Sc to Zn) all reside in the same plane (Figure 1). The bond length decreases from Sc to Co, the metal-N bond length from 1.837–2.110 Å, and after that, the metal-N bond length increments to a Zn-N length of 2.032 Å. Two of the monolayers (V and Mn), according to our calculations, exhibit differing degrees of spin polarization in their ground states, which lowers their energy below that of the non-magnetic state. The magnetic moment with the highest value is 3.22 mB for Mn, as indicated in Table 1. Additionally, we studied the Hirshfeld charge of these monolayers. As can be seen in Table 1, the nitrogen atoms have a small negative charge, while the 10 metal atoms have a small positive charge. Ionic and covalent bonds exist between the metal atoms and the atoms in their vicinity.

### 3.2. Stabilization of TM-HAB Monolayer

The formation energy represents the complexity of the catalyst preparation process. If the formation energy is negative, the preparation process will be exothermic. As a result, the lower the formation energy, the easier the preparation and the more stable the material. Figure 2 demonstrates that the formation energies of these materials are all negative, ranging from 2.59 to 7.88 eV (for further information, see Appendix A). This suggests that the ten TM-HAB single-atom catalysts we investigated should be quite simple to manufacture experimentally.

The stability of these materials is a key feature to consider when evaluating their catalytic performance. The higher the binding energy of the transition metal atom (TM) to the substrate (HAB) in TM-HAB catalysts, the better the catalytic stability. If the binding energy is too low, the metal atoms may agglomerate into clusters, reducing the catalytic effectiveness of the single-atom catalyst. According to Figure 2, the binding energies of the ten 3d transition metal elements with HAB are all negative, with Ti-HAB having the greatest negative binding energy of 13.31 eV and Zn-HAB having a lower negative binding energy of 5.38 eV. These values are also mentioned in Appendix A.

Additionally, we looked at the cohesion energy of bulk metals. Appendix A demonstrates that the cohesion energy of bulk metals, which ranges from 1.05 to 6.58 eV for Sc to Zn, is negative. If the metal atoms are likely to form clusters or are stably implanted in the skeletal layer as single atoms, it may be determined by comparing the cohesion energy and binding energy. The ten 3d transition elements from Sc to Zn that are bound to HAB have binding energies that are all lower than the cohesion energy, as shown in Figure 2. This suggests that the metal atoms can be firmly buried in the HAB monolayer as active sites.

### 3.3. The First Hydrogenation Step: Selectivity for CO_2_RR vs. HER

Each stage of the whole electrocatalytic CO_2_ reduction process calls for the participation of a proton–electron pair ($H^{+}+e^{-}$). Depending on the location of the H addition, two distinct intermediates are produced during the initial protonation of the CO_2_ reduction. Oxygen atoms may be transformed into the intermediate *COOH by adding H to them. However, the intermediate *OCHO will be created if H is added to the carbon atom. It is also possible to combine H with metal atoms to create adsorbed *H, which will cause the hydrogen evolution process. Due to the competing nature of the two CO_2_RR and HER responses, this final instance is not what we would want to observe for the CO_2_RR. Therefore, we must take into account the CO_2_RR catalysts’ ability to prevent hydrogen evolution. These steps are detailed in the equations below:(4)$$\begin{array}{c} {}*+{CO}_{2}+H^{+}+e^{-}\to^{*}COOH\\ {}*+{CO}_{2}+H^{+}+e^{-}\to^{*}OCHO\\ {}*+H^{+}+e^{-}\to^{*}H \end{array}$$

Figure 3 compares the Gibbs free energy changes for the first protonation reaction steps for the formation of *OCHO, *COOH, and *H. The detailed values are shown in Appendix A. As shown in Figure 3, for the 10 transition metals, the Gibbs free energy change for the formation of *OCHO is lower than that for the formation of *COOH intermediates for the other 9 metals except metal V, indicating that the further protonation of these 9 catalysts tends to produce *OCHO intermediates more after the activation of the adsorbed CO_2_ molecules. If $\Delta G_{{[}^{*}\mathrm{COOH}]}$ or $\Delta G_{{[}^{*}\mathrm{OCHO}]}$ are smaller than $\Delta G_{{[}^{*}H]}$, it is easier to form *COOH or *OCHO than H*. Once the active site is occupied by *COOH or *OCHO, there are few remaining active sites to form *H, so the hydrogen evolution reaction is inhibited. Therefore, the catalysts above the dashed line in Figure 3 will be dominated by the hydrogen evolution reaction, while the catalysts below the dashed line will be dominated by the desired CO_2_ reduction reaction. As can be seen from Figure 3, the Gibbs free energy change for the formation of *COOH or *OCHO is lower than that for the formation of *H. All 10 metal values are below the dashed line, so the TM-HAB monolayer we studied will have strong hydrogen evolution reaction inhibition properties.

### 3.4. Possible Product Pathways and Adsorption Energy

Since TM-HAB electrocatalytic CO_2_ reduction is a single-atom catalytic process, it is generally believed that it is difficult to generate multi-carbon products because the single-atom catalytic process cannot achieve C-C coupling between intermediates. Therefore, theoretically, it is sufficient to predict the monoatomic catalytic CO_2_ reduction process by considering the C_1_ product. The most common C_1_ products of CO_2_ electrocatalytic reduction are CO, CH_4_, HCOOH, CH_3_OH, and HCHO. Figure 4 shows the scheme for the electrocatalytic reduction of CO_2_ to obtain the C_1_ product [61].

As can be seen in Figure 4, the reduction of CO_2_ produces CO and HCOOH as a 2e process. The reduction paths are *CO_2_ → *COOH → *CO → CO and *CO_2_ → *OCHO → *HCOOH → HCOOH. The generation of HCHO is a 4e process and the reduction path is *CO_2_ → *COOH → *CO → *CHO → *OCH_2_ → HCHO. The obtaining of CH_3_OH product is a 6e process and the reduction path is *CO_2_ → *COOH → *CO → *CHO → *OCH_2_ → *OCH_3_ → *OHCH_3_ → CH_3_OH. The most complicated is the obtaining of CH_4_ product, which is an 8e process, and there are three possible paths, which are (1) *CO_2_ → *COOH → *CO → *CHO → *C → *CH → *CH_2_ → *CH_3_ → CH_4_; (2) *CO_2_ → *COOH → *CO → *CHO → *OCH_2_ → *OCH_3_ → *OHCH_3_ → *O + CH_4_ → *OH + CH_4_ → * + H_2_O + CH_4_; and (3) *CO_2_ → *COOH → *CO → *CHO → *OCH_2_ → *OCH_3_ → *OHCH_3_ → *OH + CH_4_ → * + H_2_O + CH_4_.

Based on the complexity of the CO_2_ electrocatalytic reduction reaction pathway, in order to predict the most likely products for each catalyst, we first calculated the adsorption energy of the catalyst for the C_1_ products as shown in Table 2.

Ni-HAB is relatively large for both species’ C_1_ product energy absorption, thus the product is firmly adsorbed by the catalyst and cannot be desorbed during the catalytic process, and the whole process is poisoned and no product can be obtained; this process is considered to be catalytically inactive. Similarly, for the three single-atom catalysts, Sc, Ti, and V, they have a strong adsorption capacity for CO, HCOOH, HCHO, and CH_3_OH, resulting in the inability to desorb and obtain the product; fortunately, this is not strong for CH_4_ adsorption, thus making it possible to obtain the CH_4_ product. The possible products obtained from Cr single atoms are HCHO, CH_3_OH, and CH_4_. Mn-HAB is strong enough to adsorb only CO; thus, the possible products of Mn-HAB are HCOOH, HCOH, CH_3_OH, and CH_4_. Similarly, the possible products of Co-HAB are only HCOOH, CH_3_OH, and CH_4_. However, the single-atom catalysts of Fe, Cu, and Zn are weak to adsorb the products, and all possible products can be obtained.

### 3.5. Reaction Pathways for CO_2_ Electrochemical Reduction

#### 3.5.1. HCOOH as the Main Catalytic Product

MOF electrocatalytic CO_2_ reduction to produce a single HCOOH product is a fascinating thing. Janire et al. prepared zirconium-based MFO applied to electrocatalytic CO_2_ reduction to produce a single-product formic acid in the liquid fraction [62]. We calculated the catalytic process step diagram and found that the main product of Zn-HAB for electrocatalytic CO_2_ reduction is HCOOH. The free-energy step curve is shown in Figure 5. According to the scheme in Figure 4, after CO_2_ adsorbs Zn-HAB, the first step of protonation occurs under the action of external potential to generate *COOH or *OCHO intermediates, and it can be seen in Figure 5 that the generation of *OCHO intermediates is a Gibbs-free-energy drop process, and the reaction is easily carried out. In addition, the generation of *COOH intermediate needs to cross a very high energy barrier of 0.934 eV (see Appendix A), so the first step of protonation to generate *OCHO is dominant, and the second step of paper protonation to generate *OCHOH intermediate that occurs on this basis is also a free-energy drop process. Then, after the formation of *OCHOH, the intermediate has the possibility of a *OCHOH + H^+^ + e^−^ → *CHO/*OCH + H_2_O protonation reaction, but it needs to cross the energy barrier of 1.314/1.292 eV, relatively, and the direct desorption of *OCHOH to form the HCOOH process only needs to cross the energy barrier of 0.24 eV. Detailed values of the free energy are given in Appendix A. Therefore, this step is more inclined to be terminated by HCOOH desorption. Since the tendency to form *CHO/*OCH intermediates to compete is weak, the occurrence of multi-electron steps such as 3e and 4e proceeds with difficulty, thus making it difficult to obtain CH_3_OH, HCHO, or even CH_4_ products. In summary, the main product of Zn-HAB for electrocatalytic CO_2_ reduction is HCOOH. The reaction path is * + CO_2_ → *OCHO → *OCHOH → * + HCOOH. The rate-determining step is *OCHOH → * + HCOOH with a limiting potential of 0.24 V.

#### 3.5.2. CH_3_OH and CH_4_ Are Produced Simultaneously as the Main Reduction Products

In Table 2, it is shown that the adsorption energy of Co-HAB on CO and HCHO is too large to desorb and obtain the product. Here, we analyzed the theoretical calculation of HCOOH, CH_3_OH, and CH_4_ as products and found that the main products obtained simultaneously by this catalyst were CH_3_OH and CH_4_. Figure 6 shows the whole process.

After CO_2_ adsorption by Co-HAB, the steps of paper protonation to form *COOH or *OCHO are all free energy reduction processes, which are exothermic and occur easily. Appendix A gives the detailed protonation steps and the Gibbs free energy change of each step for the electrocatalytic reduction of CO_2_ by Co-HAB. The further protonation of *COOH to *CO is also a free energy reduction process; however, the further protonation of *OCHO to *OCHOH requires a high external energy supply. Appendix A shows that 2.63 eV is required, so this step and the subsequent pathway need not be considered. In fact, *CO is further protonated to generate *CHO/*COH, but the *COH generation step needs to overcome the energy barrier of 1.73 eV (see Appendix A), and this process need not be considered, so Figure 6 only gives the *CHO generation step and considers its subsequent protonation step. It is clear that further protonation of *CHO to generate *OCH_2_ requires overcoming an energy barrier of 0.53 eV, and the subsequent steps, whether CH_3_OH or CH_4_ intermediates, are either free energy reduction processes or require overcoming energies below 0.53 eV. Thus, CH_3_OH and CH_4_ can both be generated if 0.53 eV energy is obtained from outside.

In the 6e conversion process, the formation of *CH_3_OH from *OCH_3_ intermediates is a free energy reduction process, while the formation of *O intermediates requires external energy; thus, the reaction pathway prefers the formation of *CH_3_OH. The subsequent CH_3_OH desorption process needs to overcome an energy barrier of 0.465 eV (see Appendix A), which is lower than the 0.53 eV of the *OCH_2_ formation step, and therefore the reaction pathway at 0.53 eV driven by external energy can occur smoothly. In addition, further protonation of *CH_3_OH to form *OH + CH_4_ is a free energy reduction process, and the next 8e process to generate H_2_O needs to cross the energy barrier of 0.095 eV. In summary, the main products of Co-HAB electrocatalytic CO_2_ reduction are CH_4_ and CH_3_OH, and the reaction path is * + CO_2_ → *COOH → *CO → *CHO → *OCH_2_ → *OCH_3_ → *O/*CH_3_OH → (* + CH_3_OH)/(*OH + CH4) → * + H_2_O + CH_4_. The rate-determining steps are *CHO + H_2_O + H^+^ + e^−^ → *OCH_2_ + H_2_O with a limiting potential of 0.53 V.

#### 3.5.3. HCHO, CH_3_OH, and CH_4_ Are Produced Simultaneously as the Main Reduction Products

From the adsorption calculations, it is clear that the adsorption energy of Mn-HAB is too large only for CO to obtain the C_1_ product. The adsorption energy of Fe-HAB is not large for all the five C_1_ products. Our calculations show that HCHO, CH_3_OH, and CH_4_ can be obtained with both Mn-HAB and Fe-HAB catalysts. Figure 7 shows the free energy steps of electrocatalytic CO_2_ reduction by Mn-HAB and Fe-HAB.

For the electrocatalytic CO_2_ process with both Mn-HAB and Fe-HAB catalysts, the first step of the protonation process after the activation of CO_2_ by adsorption is a free energy reduction process, whether or not *COOH or *OCHO intermediate is generated. In the 2e conversion process, the generation of *CO or *OCHOH from Mn-HAB is a free energy reduction process (see Figure 7a), while the generation of *OCHOH intermediate from Fe-HAB needs to overcome the energy barrier of 0.51 eV (see Figure 7b and Appendix A), and thus its subsequent protonation step need not be considered. In the subsequent 3e process, the lowest energy barrier step for both catalysts is *CO + H_2_O + H^+^ + e^−^ → *CHO + H_2_O, with 0.27 eV for Mn-HAB and 0.1 eV for Fe-HAB. However, in the subsequent 4e electron process, Fe-HAB also needs to overcome an energy barrier of 0.27 eV to produce the *OCH_2_ intermediate. In the further protonation process, Mn-HAB is almost always a free energy step-down process, although the final steps of HCHO, CH_3_OH desorption, and CH_4_ generation step up and need to overcome the energy barriers, but their values are 0.23 eV, 0.27 eV, and 0.14 eV, respectively (Appendix A). For 0.1 eV energy, all three products are accessible to both catalysts at an external energy of 0.27 eV. Fe-HAB is also a free energy reduction process in the subsequent protonation process, except that the HCHO and CH_3_OH desorption steps need to cross energy barriers of 0.102 eV and 0.098 eV (Appendix A), respectively. In conclusion, the Mn-HAB and Fe-HAB electrocatalytic CO_2_ reduction reactions are similar with the following pathways: * + CO_2_ → *COOH → *CO → *CHO → *OCH_2_ → *OCH_3_/(* + HCHO) → *O/*CH_3_OH → (* + CH_3_OH)/(*OH + CH_4_) → * + H_2_O + CH4. The limiting potential is 0.27 V. The difference is that the rate-determining step of Mn-HAB is *CO + H_2_O + H^+^ + e^−^ → *CHO + H_2_O, while the rate-determining step of Fe-HAB is *CHO + H_2_O + H^+^ + e^−^ → *OCH_2_ + H_2_O.

#### 3.5.4. CH_4_ as the Main Catalytic Product

Since Sc-HAB (Appendix A), Ti-HAB, and V-HAB only have a small adsorption energy for CH_4_, the other four C_1_ products will be poisoned during the catalytic process and no product will be obtained. Thus, they can only obtain a single product of CH_4_ for electrocatalytic CO_2_ reduction. In fact, the CO, HCOOH, HCHO, and CH_3_OH desorption steps in the Cu-HAB catalytic process need to overcome 0.86 eV, 1.22 eV, 0.31 eV, and 0.7 eV energy barriers, respectively (see Appendix A), all of which are higher than the energy barriers required for the CH_4_ generation step, and thus CH_4_ is considered the most likely product to be obtained. In terms of adsorption energy, HCHO, CH_3_OH, and CH_4_ can all be desorbed on the Cr-HAB surface to obtain the products, but in the actual catalytic process, the HCHO and CH_3_OH desorption steps need to overcome 0.56 eV and 0.80 eV energy barriers, respectively (see Appendix A), which are higher than the energy barriers required for the CH_4_ generation step, as shown in Figure 8d. Thus, CH_4_ is also considered to be the most likely product to be obtained.

Figure 9 shows the step diagram of Ti-HAB electrocatalytic CO_2_ reduction along the possible pathways of reduction, and the reaction equations and Gibbs free energy changes for each protonation step are shown in Appendix A. It is clear that in the 3e process, the protonation from *CO to *COH step needs to overcome the energy barrier. Appendix A shows the value of 1.12 eV for this process, but the re-protonation of *COH to *C needs to overcome the 2.12 eV energy barrier (see Appendix A). Although the free energy of the *OCHO intermediate is reduced and can be easily generated, the energy barriers required for the further protonation of *OCHO are all higher. Therefore, the reaction path for Ti-HAB electrocatalytic CO_2_ reduction is *CO_2_ → *COOH → *CO → *CHO → *OCH_2_ → *OCH_3_ → *O/*CH_3_OH → *OH + CH_4_ → * + H_2_O + CH_4_. The reaction rate-determining step is *OH + CH_4_ + H_2_O + H^+^ + e^−^ → * + CH_4_ + 2H_2_O with a limiting potential of 1.14 V.

Figure 8b shows the step diagram of V-HAB electrocatalytic CO_2_ reduction along the possible pathway reduction, and the reaction equations of each protonation step and Gibbs free energy change are shown in Appendix A. Both the first and second protonation steps are free energy reduction processes, and the *CO/*OCHOH intermediate is easily obtained. However, in the further protonation process, only the *OCHOH → *OCH step requires the lowest energy barrier to be crossed, which is 0.27 eV, and this energy barrier is the highest energy barrier for CH_4_ production and thus the limiting potential. The energy barriers to be crossed in the *CO → *COH/*CHO step are 1.523 eV and 0.453 eV, respectively (Appendix A), and the energy barrier to be crossed in the *OCHOH → *CHO step is 1.205 eV (Appendix A), which are all higher than 0.27 eV and are therefore not considered. Therefore, the pathway of V-HAB electrocatalytic CO_2_ reduction is *CO_2_ → *OCHO → *OCHOH → *OCH → *OCH_2_ → *OCH_3_ → *O/*CH_3_OH → *OH + CH_4_ → * + H_2_O + CH_4_. The rate-determining step is *OCHOH + H^+^ + e^−^ → *OCH + H_2_O, corresponding to a limiting potential of 0.27 V.

Sc-HAB, Cu-HAB, and Cr-HAB are similar to Ti-HAB in that the first step of the protonation process to generate *COOH/*OCHO after CO_2_ activation by adsorption is exothermic, but the further protonation of *OCHOH to generate *CHO/*OCH needs to overcome a higher energy barrier, while the step of *CO intermediate to generate *COH also needs to overcome a higher energy barrier. Figure 8a,c,d represent the free energy changes of intermediates of the catalytic processes of Sc-HAB, Cu-HAB, and Cr-HAB, respectively. These three catalysts have the same reaction path within the 3e process as *CO_2_ → *COOH → *CO → *CHO. However, for the Cu-HAB catalyst, the energy barrier of 1.17 eV (see Appendix A) is required to overcome the *OCH_3_ + H_2_O + H^+^ + e^−^ → *O + CH_4_ + H_2_O step in the 6th protonation step, and thus this reaction is not considered. Otherwise, the reaction paths of these three catalysts can be expressed as *CO_2_ → *COOH → *CO → *CHO → *OCH_2_ → *OCH_3_ → *CH_3_OH/*O (except Cu) → *OH + CH_4_ → * + H_2_O + CH_4_. Among them, the rate-determining step of the Sc-HAB catalytic process is *OCH_2_ + H_2_O + H^+^ + e^−^ → *OCH_3_ + H_2_O with a limiting potential of 0.29 eV; the rate-determining step of the Cu-HAB catalytic process is *CH_3_OH + H_2_O + H^+^+ e^−^ → *OH + CH_4_ + H_2_O with a limiting potential of 0.18 eV; and the rate-determining step of the Cr-HAB catalytic process is *OH + CH_4_ + H_2_O + H^+^ + e^−^ → * + CH_4_ + 2 H_2_O with a limiting potential of 0.49 eV. The rate-determining step of the Cr-HAB process is *OH + CH_4_ + H_2_O + H^+^ + e^−^ → * + CH_4_ + 2 H_2_O with a limiting potential of 0.49 eV.

### 3.6. Electronic Structure

In Section 3.5, after the Gibbs free energy change for each intermediate step, we discussed the rate-determining step, the limiting potential, and the corresponding major products for each catalyst as shown in Table 3. In summary, all catalysts except Ni-HAB are electrocatalytically active for CO_2_ reduction, and the main products of Sc, Ti, V, Cr, and Cu are CH_4_. The limiting potential and overpotential of Ti-HAB are the highest, at 1.14 V and 1.31 V, respectively, and the overpotentials of the other monolayer catalysts are in the range 0.01–0.7 V, which are comparable to those of the most active step surface Cu(211) (η = 0.77 V) and the overpotential of the most active metal surface Pt(111) (η = 0.46 V) [63]. In addition, our results were compared with experimentally prepared catalysts. An example is the work of Xu’s team, who synthesized non-peripheral octamethyl-substituted cobalt(II) phthalocyanine (N-CoMePc) catalysts, which achieved a Faraday efficiency of up to 94.1% for CO production at a low overpotential of 0.6 V [64]. Another typical example is the work of Ivan et al. [65], which involves the synthesis, description, and preliminary evaluation of bimetallic copper-based hollow fiber electrodes with a compact three-dimensional geometry to overcome mass transfer limitations and improve the electrochemical conversion of CO. It is noted that the generation of CO occurs in the range 1–1.5 V. Some of our predicted TM-HABs have comparable or even lower overpotentials than N-CoMePc and bimetallic Cu-based hollow fiber. Thus, our theoretical results suggest a very promising single-atom catalyst for electrocatalytic CO_2_ reduction.

The metal-ligand bonding theory of organometallic catalysts reveals that the interactions between catalysts and intermediates are mainly σ- and π-bonds. Figure 10 shows a clear overlap between the 3d orbital of the metal atom and the 2p orbital of the O atom or C atom in the decisive step intermediates (*OCH_2_, O*H, O*CHOH, *CO, *CHO, or *OHCH_3_), either spin-up or spin-down, which indicates a good interaction between the TM-HAB and the intermediate. However, the overlap effect of 3d and 2p orbitals in Figure 10b is better than the overlap of 3d and 2p orbitals in Figure 10a,c–i. This indicates that the interaction of Ti-HAB with the corresponding intermediates is stronger than that of the other catalysts. In addition, it can be seen from Table 3 that the limiting potential U_L_ of −1.14 eV for the electrocatalytic CO_2_ reduction by Ti-HAB is larger than that of the other catalysts. This is in good agreement with the results of PDOS. The stronger the interaction, the more stable the adsorption intermediate system, and the higher energy barrier that needs to be overcome to ensure that the catalytic reaction occurs, which leads to a larger increase in the free energy of the Ti-HAB catalyzed CO_2_ reduction decisive step, and thus a more negative limiting potential for the reduction reaction.

## 4. Conclusions

In summary, we investigated the electrocatalytic CO_2_ reduction reaction of single-atom catalysts created from transition metal-hexaaminobenzene two-dimensional coordination network materials. Density functional theory calculations show that for 10 transition metal TM-HAB monolayers ranging from Sc to Zn, the binding energy of the metal atoms to the HAB is large enough for the metal atoms to be stably dispersed in the HAB monolayer. All of these materials inhibit the hydrogen evolution reaction. The reduction products of Sc, Ti, V, Cr, and Cu are mainly CH_4_. The reduction products of Zn are mainly HCOOH. The main reduction products of Co are CH_3_OH and CH_4_. The reduction products of Mn and Fe are mainly HCHO, CH_3_OH, and CH_4_. Among them, the limiting potential of Ti-HAB is 1.14 eV and the overpotential is 1.31 V. The overpotentials of the other monolayers were in the range 0.01–0.7 V. All of the values are relatively low; therefore, we predict that a TM-HAB monolayer will exhibit strong catalytic activity in the electrocatalytic reduction of CO_2_, making it a promising electrocatalyst for CO_2_ reduction.

## Figures and Tables

**Figure 1 nanomaterials-12-04005-f001:**
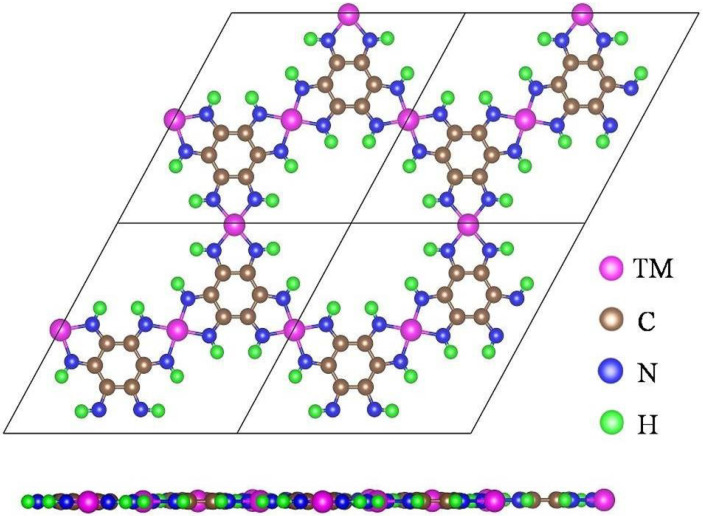
The top (**up**) and side (**down**) views of the structures in a 2 × 2 supercell for ten TM-HAB monolayers.

**Figure 2 nanomaterials-12-04005-f002:**
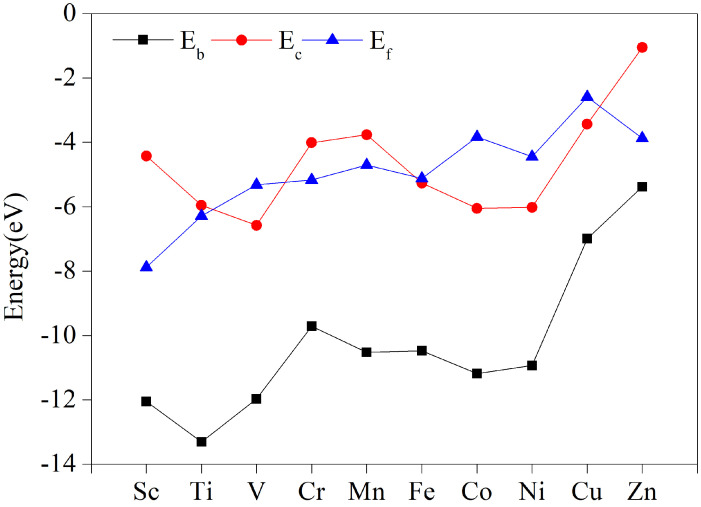
Stability of different TM-HAB monolayers structurally. TM is the transition metal atom, E_c_ is the cohesive energy of the TM bulk, E_b_ is the binding energy between TM and HAB in TM-HAB, and E_f_ is the formation energy of TM-HAB.

**Figure 3 nanomaterials-12-04005-f003:**
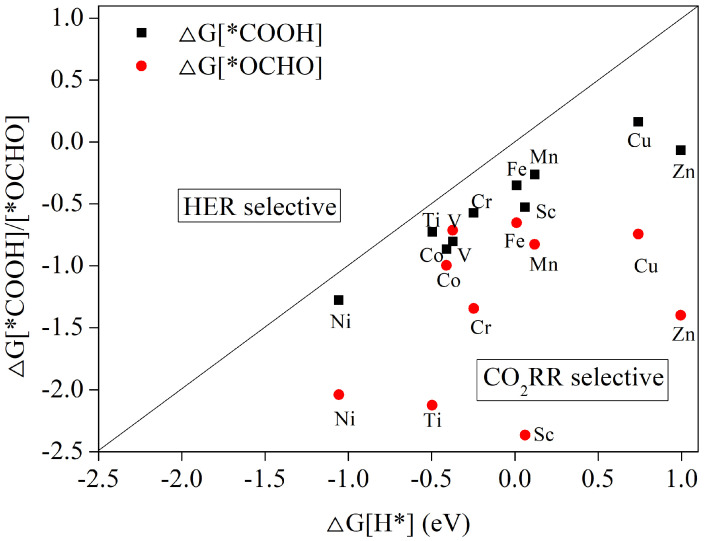
The Gibbs free energy change of the first protonation step in the CO_2_ reduction reaction (CO_2_RR) and H_2_ evolution reaction (HER). CO_2_RR selective catalysts are those located below the dotted line.

**Figure 4 nanomaterials-12-04005-f004:**
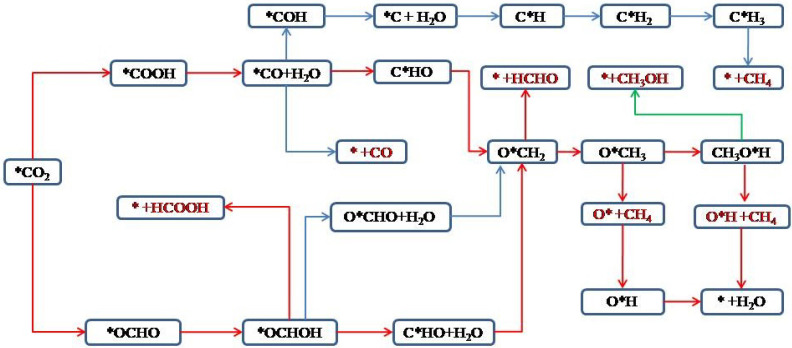
Flow chart of electrocatalytic CO_2_ reduction to C_1_ product scheme; red is the final product [61].

**Figure 5 nanomaterials-12-04005-f005:**
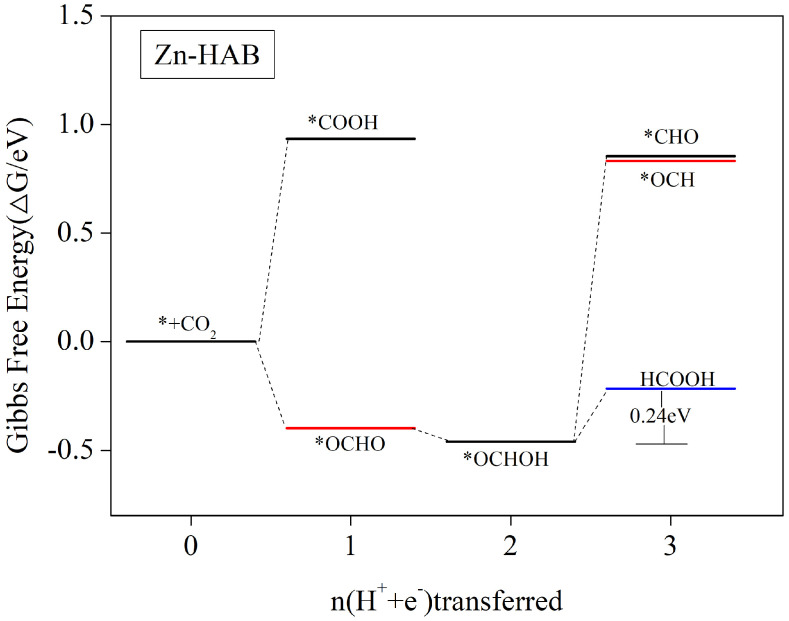
At zero potential, Gibbs free energy profiles for the CO_2_RR along the most favorable routes for Zn-HAB. A CO_2_ molecule in the gas phase with a clean catalytic surface is assigned a free energy of zero.

**Figure 6 nanomaterials-12-04005-f006:**
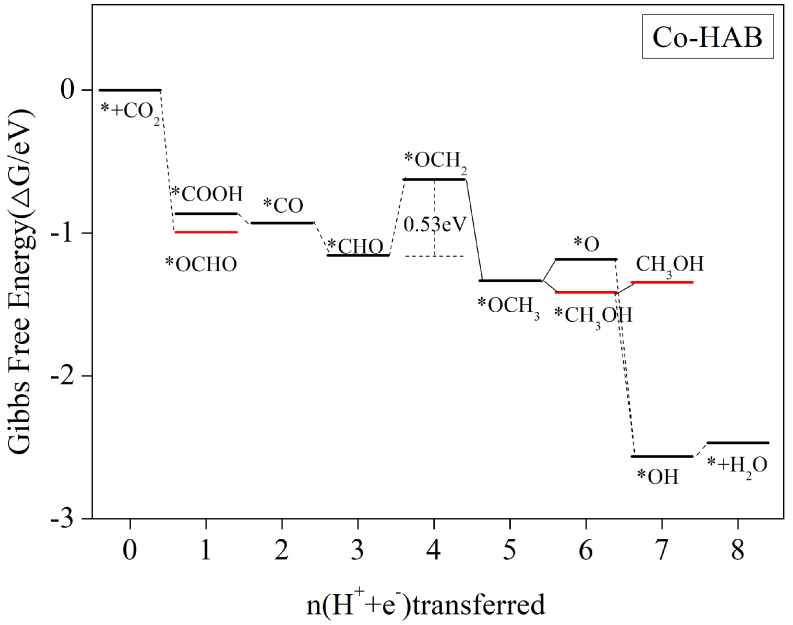
Gibbs free energy profiles for the CO_2_RR along the most favorable pathways for Co–HAB at zero potential. The free energy zero is set to a CO_2_ molecule in the gas phase with a clean catalyst surface.

**Figure 7 nanomaterials-12-04005-f007:**
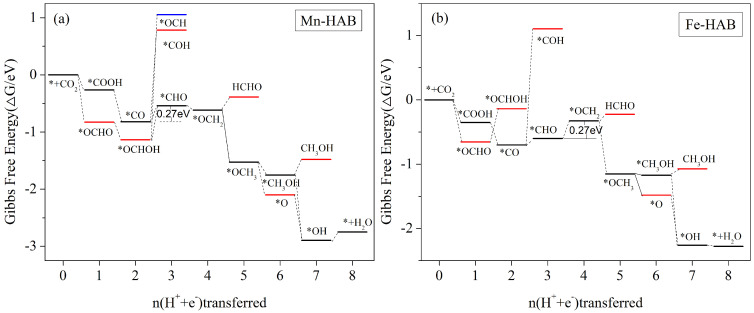
Gibbs free energy curves for (**a**) Mn-HAB and (**b**) Fe-HAB, at zero potential, along the most favorable path of the CO_2_RR. The free energy zero point was set as the carbon dioxide molecule in the gas phase with a clean catalyst surface.

**Figure 8 nanomaterials-12-04005-f008:**
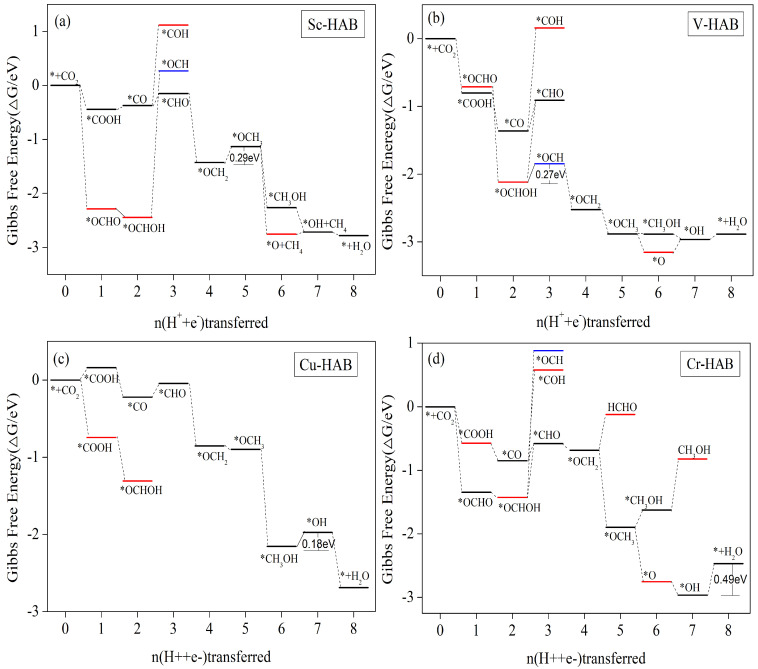
Gibbs free energy profiles for the CRR along the most favorable pathways for (**a**) Sc–HAB, (**b**) V–HAB, (**c**) Cu–HAB, and (**d**) Cr–HAB at zero potential. The free energy zero is set to a CO_2_ molecule in the gas phase with a clean catalyst surface.

**Figure 9 nanomaterials-12-04005-f009:**
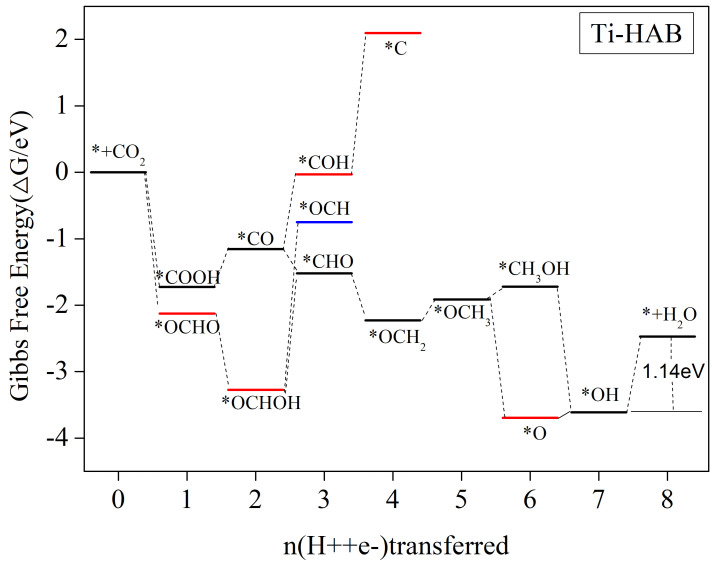
Gibbs free energy curve of CO_2_RR along the most favored path of Ti-HAB at zero potential. The free energy zero has been set as the free energy of CO_2_ molecules in the gas phase with a clean catalyst surface.

**Figure 10 nanomaterials-12-04005-f010:**
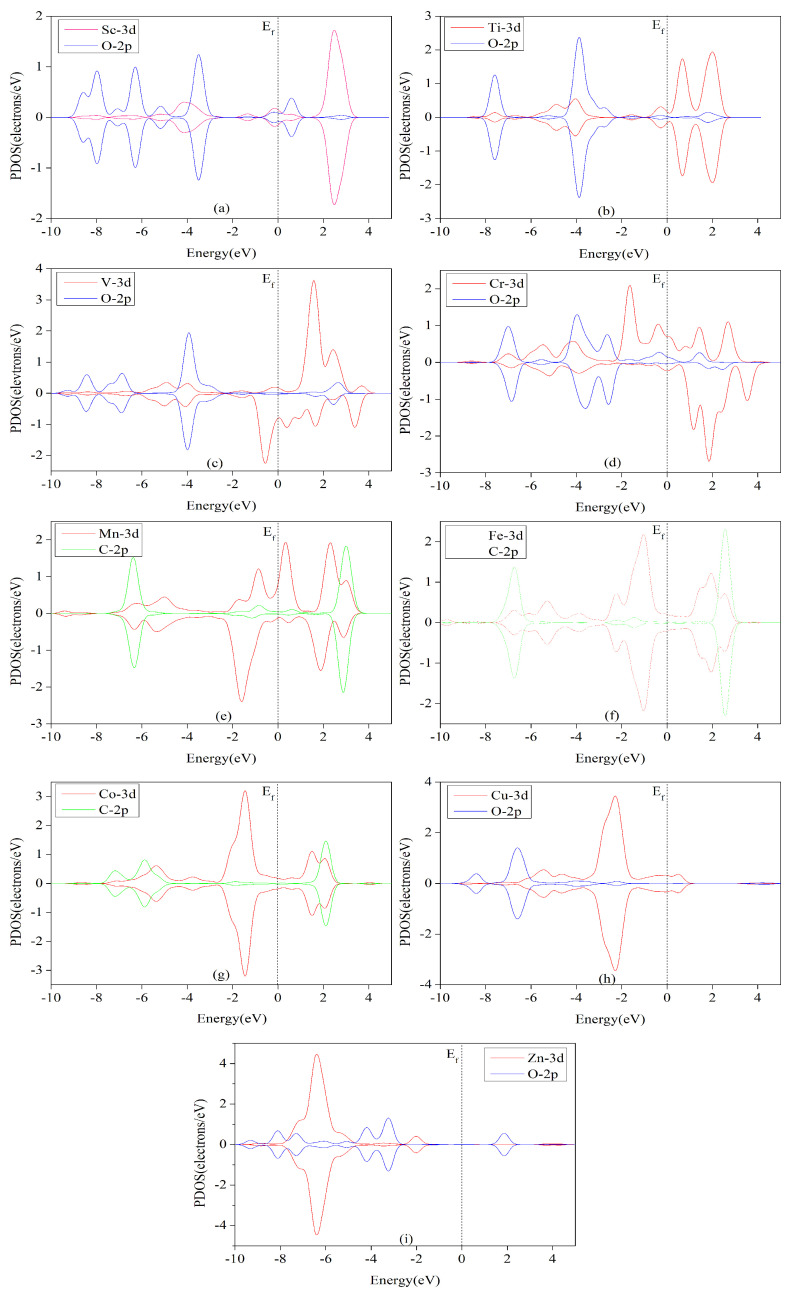
The projected partial density of states of *OCH_2_ adsorbed on Sc, *OH adsorbed on Ti and Cr, O*CHOH adsorbed on V and Zn, *CO adsorbed on Mn and Fe, *CHO adsorbed on Co, and *OHCH_3_ adsorbed on Cu. The dotted lines denote the Fermi level. The red, green, and blue lines represent the 3d orbital of the metal atoms, the 2p orbital of the oxygen atoms, and the 2p orbital of the carbon atoms, respectively.

**Table 1 nanomaterials-12-04005-t001:** Hirshfeld charge on metal atoms (Q_TM_) and on nitrogen atoms (Q_N_) for the ten TM-HAB model systems. The Hirshfeld spin of the metal atoms and the M-N bond length (R_M−N_) are shown.

TM-HAB	Q_TM_	Spin-TM	QN/e	RTM-N/Å
Sc	0.708	0.000	−0.259	2.110
Ti	0.588	0.000	−0.236	1.962
V	0.349	−1.935	−0.202	1.959
Cr	0.446	0.000	−0.222	1.929
Mn	0.321	−3.225	−0.203	1.876
Fe	0.132	0.000	−0.162	1.924
Co	0.049	0.000	−0.153	1.837
Ni	0.052	0.000	−0.147	1.838
Cu	0.324	0.000	−0.208	1.945
Zn	0.394	0.000	−0.220	2.032

**Table 2 nanomaterials-12-04005-t002:** Adsorption energy (Eads/eV) of different CO_2_ reduction products.

TM-HAB	CO	HCOOH	HCHO	CH_3_OH	CH_4_
Sc-HAB	−1.263	−1.432	−0.998	−1.361	−0.100
Ti-HAB	−2.767	−2.013	−2.461	−2.334	−0.848
V-HAB	−1.910	−0.804	−1.296	−1.258	−0.125
Cr-HAB	−2.209	−1.558	−0.161	−0.838	−0.109
Mn-HAB	−1.617	−0.081	−0.073	−0.274	−0.243
Fe-HAB	−0.165	−0.073	−0.070	−0.123	−0.071
Co-HAB	−1.794	−0.895	−1.008	−0.781	−0.525
Ni-HAB	−2.309	−1.994	−1.825	−2.275	−2.044
Cu-HAB	−0.547	0.053	0.068	−0.473	−0.553
Zn-HAB	−0.218	−0.109	−0.151	−0.185	−0.106

**Table 3 nanomaterials-12-04005-t003:** Calculated potential-determining steps (PDS) for Sc-Zn (except Ni) for nine materials, the limiting potential (U_L_) for the CO_2_RR reaction, and the corresponding major products.

TM-HAB	PDS	U_L_/V	Main Products and Corresponding Overpotentials (η/V)
Sc-HAB	*OCH_2_ + H_2_O + H^+^ + e^−^ → *OCH_3_ + H_2_O	−0.29	CH_4_(0.46)
Ti-HAB	*OH + CH_4_ + H_2_O + H^+^ + e^−^ → * + CH_4_ + 2H_2_O	−1.14	CH_4_(1.31)
V-HAB	*OCHOH + H^+^ + e^−^ → *OCH + H_2_O	−0.27	CH_4_(0.44)
Cr-HAB	*OH + CH_4_ + H_2_O + H^+^ + e^−^ → * + CH_4_ + 2H_2_O	−0.27	CH_4_(0.66)
Mn-HAB	*CO + H_2_O + H^+^ + e^−^ → *CHO + 2H_2_O	−0.27	HCHO(0.2),CH_3_OH(0.29),CH_4_(0.44)
Fe-HAB	*CO + H_2_O + H^+^ + e^−^ → *CHO + 2H_2_O	−0.27	HCHO(0.2),CH_3_OH(0.29),CH_4_(0.44)
Co-HAB	*CHO + H_2_O + H^+^ + e^−^ → *OCH_2_ + 2H_2_O	−0.53	CH_3_OH(0.55),CH_4_(0.70)
Cu-HAB	*CH_3_OH + H_2_O + H^+^ + e^−^ → *OH + CH_4_ + H_2_O	−0.18	CH_4_(0.35)
Zn-HAB	*OCHOH → * + HCOOH	0.24	HCOOH(0.01)

## Data Availability

The data presented in this study are available in Appendix A.

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
