# Peer review of "Two-Dimensional Transition Metal-Hexaaminobenzene Monolayer Single-Atom Catalyst for Electrocatalytic Carbon Dioxide Reduction"

_nanomaterials, 2022, doi:10.3390/nano12224005_

Round 1
Reviewer 1 Report
This manuscript reports the theoretical investigation of the two-dimensional single-atom catalysts for electrochemical carbon dioxide reduction reaction (CO2RR). The authors adopted transition metal-hexaaminobenzene (TM-HAB) monolayers as the model structure with different TM catalytic active centers. I would support the publication of this manuscript after the following points are well-addressed.
(1) In general, compared with two-electron products (CO, HCOOH, and H2), the theoretical calculation results for hydrocarbon products (CH4, C2H4, and C2H5OH) are not well-consistent with the experimental results, which originate from the fact that their production accompanies many-electron transfer process involving multiple reaction intermediates. To reflect on this point, the authors should expand their discussion.
(2) TM-coordinated molecular catalysts such as porphyrin and phthalocyanine have been investigated as CO2RR catalysts mainly owing to their well-defined TM sites. The authors must compare the current results with the previous works (see below). Especially, many experimental data suggest that Co-based phthalocyanine catalysts are selective toward CO with high Faradaic efficiency values (>95 %).
(3) The DFT model should be realistic with the experimental cases. How real the TM-HAB catalysts are? Is there any experimental work?
(4) Overall, this manuscript should provide the full name of acronyms before their first appearance.
[Refs in Q2]
10.1038/ncomms14675, 10.1021/acscatal.7b02220, 10.1021/jacs.1c02326, 10.1021/jacs.2c06953, 10.1021/jacs.2c09862, 10.1016/j.carbon.2020.06.036,
Author Response
Dear reviewer,
We would like to thank the comments from you, which much helpful for revision of our manuscript (#nanomaterials-2002097). We revised our manuscript accordingly and the corrections are listed below:
- In general, compared with two-electron products (CO, HCOOH, and H2), the theoretical calculation results for hydrocarbon products (CH4, C2H4, and C2H5OH) are not well-consistent with the experimental results, which originate from the fact that their production accompanies many-electron transfer process involving multiple reaction intermediates. To reflect on this point, the authors should expand their discussion.
Thank you very much for your suggestion. We have extended the discussion. The four catalysts, Zn, Co, Fe, and V, are supplemented with a discussion of the possibility of subsequent multi-electron processes. Also, the reaction pathways, rate determining steps, and limiting potentials of Zn, Co, Mn, Fe, Ti, and V electrocatalytic CO2 reduction are added in detail.
- TM-coordinated molecular catalysts such as porphyrin and phthalocyanine have been investigated as CO2RR catalysts mainly owing to their well-defined TM sites. The authors must compare the current results with the previous works (see below). Especially, many experimental data suggest that Co-based phthalocyanine catalysts are selective toward CO with high Faradaic efficiency values (>95 %).
Thank you very much for your suggestion. We have extended and compared the non-peripheral octamethyl-substituted cobalt(II) phthalocyanine (N-CoMePc) and bimetallic Cu-based hollow fibre catalysts by using the states “In addition, our results are compared with experimentally prepared catalysts. An example is the work of Xu's team, who synthesized non-peripheral octamethyl-substituted cobalt(II) phthalocyanine (N-CoMePc) catalysts, which achieved a Faraday efficiency of up to 94.1% for CO production at a low overpotential of 0.6 V. Another typical example is Ivan et al. Synthesis, description and preliminary evaluation of bimetallic copper-based hollow fiber electrodes with a compact three-dimensional geometry to overcome mass transfer limitations and improve the electrochemical conversion of CO.It is noted that the generation of CO in the range of 1V-1.5V . Some of our predicted TM-HABs have comparable or even lower overpotentials than N-CoMePc and bimetallic Cu-based hollow fibre.”
- The DFT model should be realistic with the experimental cases. How real the TM-HAB catalysts are? Is there any experimental work?
Thank you very much for your suggestion. We have added the arguments for TM-HAB as a potential catalyst, citing also the report of Ni-HAB catalyzed ORR. The details are as follows:“The low-coordinated TM atoms are stably anchored as metal centers in the MOFs TM3(HAB)2, exhibiting the characteristics of single-atom catalysts (SACs). Therefore, they can be considered as SACs with high practicality. The metal center in the two-dimensional TM-HAB plays the role of an active site with catalytic properties, and the type of central atom can be tuned to meet the catalytic requirements. Notably, TM-HAB is formed by each TM atom with four surrounding N. The TMN4 complex is an analogue of TMNx, which exhibits good catalytic properties similar to those of noble metals in various catalysts. such as oxygen reduction, nitrogen fixation and carbon dioxide reduction. Park et al. proposed a two-dimensional (2D) conductive metal-organic framework consisting of M-N4 units (M = Ni, Cu) and hexaaminobenzene (HAB) linkers as a catalyst for oxygen reduction reactions, and the results showed that the catalytic performance depends strongly on the metal species. However, the application of TM-HAB monolayers for CO2 reduction has been little reported so far. This motivated us to explore whether TM-HAB constructed with different metal species could be used as prospective electrocatalysts for CO2 reduction.”
- Overall, this manuscript should provide the full name of acronyms before their first appearance.
Thank you very much for your suggestion. We have replaced TM-HAB with transitionmetal-hexaaminobenzene (TM-HAB) by the full name description of TM-HAB in the title.
5 [Refs in Q2]
10.1038/ncomms14675, 10.1021/acscatal.7b02220, 10.1021/jacs.1c02326, 10.1021/jacs.2c06953, 10.1021/jacs.2c09862, 10.1016/j.carbon.2020.06.036,
Thank you very much for your suggestion. We have added DOI numbers to all references.
Thank you again for your positive comments and valuable suggestions to improve the quality of our manuscript.
Best wishes.
Xianshi Zeng
Nanchang University

Reviewer 2 Report
The electrochemical reduction of CO2 is one of those research areas that deserve attention nowadays and this report examine the catalytic activity of TM-HAB monolayers including different metals for CO2 conversion to CH4, CH3OH, and others. The operating principles of the analyses are clear. Technically speaking, the report is valuable and it will stimulate researchers in the field, although the authors need to further compare their study with previous research on the topic. Thus, I believe the manuscript might be published in Nanomaterials after revision. These are the comments from my side:
- There are previous reports on the use of single atoms from TM-HAB and the authors should make an effort to further emphasize the contribution provided by this work. If not, the work done could somehow be seen as incremental.
- The authors need also to consider previous reports on the use of Cu-based electrocatalysts for CO2 reduction to alcohols and hydrocarbons, including CH4, e.g.,
Catal. Today, 346, 2020, 34-39;
Micropor. Mesopor. Mat., 284, 2019, 128-132;
I believe these reports would be of interest to enhance the discussion in the present report. It would also give a more comprehensive literature review.
Author Response
Dear reviewer,
We would like to thank the comments from you, which much helpful for revision of our manuscript (#nanomaterials-2002097). We revised our manuscript accordingly and the corrections are listed below:
- There are previous reports on the use of single atoms from TM-HAB and the authors should make an effort to further emphasize the contribution provided by this work. If not, the work done could somehow be seen as incremental.
Thank you very much for your suggestion. We have added the arguments for TM-HAB as a potential catalyst, citing also the report of Ni-HAB catalyzed ORR. The details are as follows:“The low-coordinated TM atoms are stably anchored as metal centers in the MOFs TM3(HAB)2, exhibiting the characteristics of single-atom catalysts (SACs). Therefore, they can be considered as SACs with high practicality. The metal center in the two-dimensional TM-HAB plays the role of an active site with catalytic properties, and the type of central atom can be tuned to meet the catalytic requirements. Notably, TM-HAB is formed by each TM atom with four surrounding N. The TMN4 complex is an analogue of TMNx, which exhibits good catalytic properties similar to those of noble metals in various catalysts. such as oxygen reduction, nitrogen fixation and carbon dioxide reduction. Park et al. proposed a two-dimensional (2D) conductive metal-organic framework consisting of M-N4 units (M = Ni, Cu) and hexaaminobenzene (HAB) linkers as a catalyst for oxygen reduction reactions, and the results showed that the catalytic performance depends strongly on the metal species. However, the application of TM-HAB monolayers for CO2 reduction has been little reported so far. This motivated us to explore whether TM-HAB constructed with different metal species could be used as prospective electrocatalysts for CO2 reduction.”
- The authors need also to consider previous reports on the use of Cu-based electrocatalysts for CO2 reduction to alcohols and hydrocarbons, including CH4, e.g.,
Catal. Today, 346, 2020, 34-39;
Micropor. Mesopor. Mat., 284, 2019, 128-132;
I believe these reports would be of interest to enhance the discussion in the present report. It would also give a more comprehensive literature review.
Thank you very much for your suggestion. We have extended and compared the non-peripheral octamethyl-substituted cobalt(II) phthalocyanine (N-CoMePc) and bimetallic Cu-based hollow fibre catalysts by using the states “In addition, our results are compared with experimentally prepared catalysts. An example is the work of Xu's team, who synthesized non-peripheral octamethyl-substituted cobalt(II) phthalocyanine (N-CoMePc) catalysts, which achieved a Faraday efficiency of up to 94.1% for CO production at a low overpotential of 0.6 V. Another typical example is Ivan et al. Synthesis, description and preliminary evaluation of bimetallic copper-based hollow fiber electrodes with a compact three-dimensional geometry to overcome mass transfer limitations and improve the electrochemical conversion of CO.It is noted that the generation of CO in the range of 1V-1.5V . Some of our predicted TM-HABs have comparable or even lower overpotentials than N-CoMePc and bimetallic Cu-based hollow fibre.”
We have added content to the discussion citing Micropor. Mesopor. Mat., 284, 2019, 128-132; which reads as follows: MOF electrocatalytic CO2 reduction to produce a single HCOOH product is a fascinating thing.Janire et al. prepared zirconium-based MFO applied to electrocatalytic CO2 reduction to produce a single product formic acid in the liquid fraction.
Thank you again for your positive comments and valuable suggestions to improve the quality of our manuscript.
Best wishes.
Xianshi Zeng
Nanchang University

Round 2
Reviewer 2 Report
I believe all comments aroused have been tackled correctly.